

# Temporal variability in foraminiferal morphology and geochemistry at the West Antarctic Peninsula: a sediment trap study

Anna Mikis[1], Katharine R. Hendry[2], Jennifer Pike[1], Daniela N. Schmidt[2], Kirsty M. Edgar[2,3], Victoria Peck[4], Frank J.C. Peeters[5], Melanie J. Leng[6], Michael P. Meredith[4], Chloe L.C. Jones[2], Sharon Stammerjohn[7], and Hugh Ducklow[8]

[1] School of Earth and Ocean Sciences, Cardiff University, Main Building, Cardiff, CF10 3AT, UK
[2] School of Earth Sciences, University of Bristol, Wills Memorial Building, Queen's Road, Bristol, BS8 1RJ, UK
[3] now at School of Geography, Earth and Environmental Sciences, University of Birmingham, Edgbaston, Birmingham, B15 2TT, UK
[4] British Antarctic Survey, High Cross, Madingley Road, Cambridge, CB3 0ET, UK
[5] Department of Earth Sciences, Faculty of Sciences, Vrije Universiteit Amsterdam, De Boelelaan 1085, 1081 HV, Amsterdam, The Netherlands
[6] NERC Isotope Geosciences Facilities, British Geological Survey, Keyworth, Nottingham, NG12 5GG, UK and Centre for Environmental Geochemistry, School of Biosciences, Sutton Bonington Campus, University of Nottingham, Loughborough, LE12 5RD, UK
[7] Institute of Arctic and Alpine Research, University of Colorado Boulder, Boulder, CO 80303, USA
[8] Lamont-Doherty Earth Observatory, Columbia University, 61 Route 9W, Palisades, NY 10964-1000 USA

*Correspondence to*: Katharine R. Hendry (K.Hendry@bristol.ac.uk)

**Abstract.** The West Antarctic Peninsula (WAP) exhibits strong spatial and temporal oceanographic variability, resulting in highly heterogeneous biological productivity. Calcifying organisms that live in the waters off the WAP respond to temporal and spatial variations in ocean temperature and chemistry. These marine calcifiers are potentially threatened by regional climate change with waters already naturally close to carbonate undersaturation. Future projections of carbonate production in the Southern Ocean are challenging due to the lack of historical data collection and complex, decadal climate variability. Here we present a six-year long record of the shell fluxes, morphology, and stable isotope variability of the polar planktic foraminifera *Neogloboquadrina pachyderma* (sensu stricto) from near Palmer Station, Antarctica. This species is fundamental to Southern Ocean planktic carbonate production as it is one of the very few planktic foraminifer species adapted to the marine polar environments. We use these new data to obtain insights into its ecology and to derive a robust assessment of the response of this polar species to environmental change. Morphology and stable isotope composition reveal the presence of different growth stages within this tightly defined species. Inter- and intra-annual variability of foraminiferal flux and size is evident and driven by a combination of environmental forcing parameters, most importantly food availability, temperature, and sea-ice duration and extent. Foraminiferal growth occurs throughout the austral year and is influenced by environmental change, a large portion of which is driven by the Southern Annular Mode and El Niño Southern Oscillation. A distinct seasonal production is observed, with highest shell fluxes during the warmest and most productive months of the year. The sensitivity of calcifying foraminifera to environmental variability in this region, from weeks to



decades, has implications both for their response to future climatic change, and for their use as palaeoclimate indicators. A longer ice-free season could increase carbonate production in this region at least while carbonate saturation is still high enough to allow for thick tests to grow.

# 1 Introduction

The West Antarctic Peninsula (WAP; Figure 1) is a highly climatically sensitive region, characterised by strong seasonal and interannual variability in atmospheric, cryospheric and oceanographic conditions (Martinson et al., 2008). WAP marine primary production is spatially heterogeneous, and concentrated into very productive biological hotspots that feed higher trophic level organisms and rich benthic ecosystems (Grange and Smith, 2013). Biological production along the WAP responds to shifts in the seasonality of sea-ice extent and water column properties, both of which are inherently linked with
atmospheric and oceanographic changes and show significant spatial and temporal variability (Kim et al., 2018).

Planktic calcifiers are important for pelagic biogeochemical cycling along the WAP. The extreme environments of the WAP make these species particularly vulnerable to seasonal and regional changes in food availability and water chemistry as a result of variable carbonate ion saturation states and overall low saturation (Lencina-Avila et al., 2018). In addition to their ecosystem function, fossil shells of marine calcifiers from sediment cores have long been used to unravel past changes
in the marine environment, based upon the assemblage composition, shell morphology and their geochemistry (Spindler and Dieckmann, 1986). For example, the oxygen isotopic composition of WAP foraminiferal calcite shells (termed tests) can be used in the reconstruction of past changes in continental ice volume, water temperature, as well as for age estimation of sediments by stratigraphic correlation (Peck et al., 2015).

*Neogloboquadrina pachyderma* is the dominant planktic foraminifer in high latitudes (Tolderlund, 1971). Although *N.*
*pachyderma* generally grow in the open ocean at pycnocline depths, there is evidence that increase of shell mass, due to addition of a calcite crust, may occur below the mixed layer down to a depth of 200m (Kohfeld et al., 1996). *N. pachyderma* have a maximum generation time of approximately one year, which culminates in the development of a thick secondary encrustation associated with gametogenesis (Kozdon et al., 2009). The species has been observed to live in sea-ice brine channels under salinity conditions at least up to 58 on the practical salinity scale (Spindler, 1996) making its ecology unique
amongst planktic foraminifers (Spindler and Dieckmann, 1986). Due to its dominance, test size changes in this species can strongly influence carbonate export in high latitudes and accumulation in the deep ocean (Huber et al., 2000).

Understanding the impact of a changing climate on planktic foraminifera – when superimposed on the high environmental variability of the WAP – is challenging and requires the study of long-term (decadal-scale) observations. This challenge is complicated by the fact that for a long time two genotypes of the morphospecies have been considered as one
species (*N. pachyderma* and *N. incompta* – previously known as *N. pachyderma* (sinistral and dextral, respectively)) raising concerns about our historical understanding of the palaeoecology of the species and its use as a palaeo-proxy carrier. Further, along the WAP, two distinct genotypes of *N. pachyderma* 'sinistral' have been found (Darling et al., 2006) but their



ecological adaptations are not well understood due to the scarcity of plankton net and sediment trap data. In the Arctic, different morphotypes of the sinistral *N. pachyderma* (sensu stricto) have been identified from core top sediments, with distinct size distributions and geochemical signatures (Altuna et al., 2018). Studies on an Arctic genotype of *N. pachyderma* under ocean acidification conditions show reduced carbonate production moderated by warming (Manno et al., 2012)

making it difficult to predict future climate change impacts on this morphospecies. Finally, the impact of varying environmental conditions on the degree of secondary encrustation of *N. pachyderma,* and consequent impact on geochemical proxies, are poorly constrained (Kozdon et al., 2009).

To this end, we investigate the controls on *N. pachyderma* foraminiferal flux, morphology (including the existence of different life stages), and geochemistry using a long-term archive of foraminifera from a sediment trap moored near Palmer

Station, WAP. This unique observational record allows us to address the sensitivity of foraminifera to high-frequency (weeks to decades) environmental change, how oceanographic signals are recorded in exported foraminiferal tests, and implications for the use of carbonate-based palaeo-proxies in Antarctic environments.

## 2 Methods

### 2.1    Physical and biogeochemical properties and the Palmer LTER sediment trap site)

The shelf adjacent to Anvers Island, WAP, is one of the key localities of the Palmer Long Term Ecological Research program (Palmer LTER; Figure 1). A sediment trap (PARFLUX Mark 78H 21-sample trap, McLane Research Labs, Falmouth, MA) has been deployed at 170 m water depth (total water column depth 350 m, 64° 30'S, 66° 00'W) since 1993 (Gleiber et al., 2012). Sea surface temperature (SST), sea ice concentration (SIC), chlorophyll *a* (Chl *a*) concentration, and organic carbon and nitrogen fluxes for the Palmer LTER site between Jan 2006 and Jan 2013 were obtained from published

archives (NOAA NCEP, 2015; NSIDC, 2015; Palmer Station LTER Antarctica, 2015).

The sediment trap site is typically bathed by modified Circumpolar Deep Water (CDW) upwelled onto the shelf. This CDW is overlain in winter by a thick (~50-100 m) surface mixed layer of water close to the freezing point, which persists into summer as Winter Water (WW) - the cold remnant of the deep winter mixed layer. WW is capped with a thin layer of Antarctic Surface Water during stratified austral summer months, which is warmed by insolation and freshened by ice melt

(Moffat and Meredith, 2018). Primary production on the continental shelf of the WAP is primarily driven by the seasonal cycle of sea ice, light availability (insolation), and atmospheric and oceanographic circulation (Vernet et al., 2008). During November to February (austral summer), the Palmer LTER sediment trap is located within the outer marginal ice zone, where the availability of nutrients and the physical properties of the water column vary both temporally and spatially (Kim et al., 2016). This variability, in turn, leads to variations in the onset and extent of primary and export production (Huang et al.,

30  2012).

Between 2006 and 2012, SST and water temperature at 100 m depth showed pronounced seasonal and inter-annual variability (SST shown in Figure 2; Supplementary Information). There is a strong seasonal stratification of the upper water



column between late-spring and early-autumn throughout the time series, resulting in an increased difference between SST and temperature at 100 m water depth. SIC and sea-ice duration shows a clear seasonal cycle with pronounced inter-annual variability. Peak SIC occurs from July, with the exact timing and maximum concentration varying between years. It is assumed that chlorophyll *a* (Chl *a*) concentration, a proxy for algal standing stocks, is minimal during the sea ice season

(Figure 2). Peak Chl *a* concentration, which also shows strong inter-annual variability, is reached after the end of the sea ice season when glacial meltwater introduces nutrients and algal cells to the surface waters, which – together with the stratified water column – promote primary productivity, especially the development of diatom blooms. Total organic carbon (and nitrogen) flux is lowest between April and October each year (austral winter; Figure 2E) when the upper water column is well-mixed, and highest in the summer during times of maximum stratification. Variability in the organic carbon flux is also

observed between the individual years throughout the record as well as within years (Figure 2E).

## 2.2 Foraminiferal flux measurements

Sediment trap samples were stored at 5°C until processing. *Neogloboquadrina pachyderma* sensu stricto (Nps) specimens from archived splits, or whole samples of picked sediment trap material, were counted using a Zeiss Stemi 2000 optical microscope at 50x magnification. Individual Nps were picked into separate holding trays and counted using a

specimen tally counter.

## 2.3 Morphological measurements

Morphological analysis of the Nps specimens was carried out both manually and using an automated microscopy technique. The automated high-throughput method was used to determine the morphological character of the entire Nps population at the site.  The manual method was conducted on a smaller number of individuals (maximum 50) as an

independent validation of automated measurements.

### 2.3.1 Automated analysis

Automated analysis of bulk samples used an automated microscope and image analysis system (Bollmann et al., 2005), which scans and captures images via a 12 MP Olympus CC12 camera attached to a Wild MZ3 incident light microscope. Fifty-four samples were measured, with the number of specimens in each ranging from 6 to 3222. Analysis3.0

software was used to generate morphometric data on twenty-one different parameters including: area, perimeter, minimum diameter, maximum diameter, aspect ratio, elongation, sphericity and mean grey value. Particles with both a sphericity shape factor below 0.4 and grey values below 50 and above 220 were subsequently excluded from analyses as these particles were not whole foraminifera but likely broken specimens or lithogenic material. A further manual inspection of the images removed any other contaminants within the sample to avoid biasing outputs. Diagnostic features of the specimens, the mean,

minimum and maximum values for shell maximum diameter, mean grey scale and sphericity, were calculated from the remaining individuals.



### 2.3.2 Manual analysis

Approximately fifty specimens per sample were fixed onto glass slides and imaged using an Olympus SZX7 transmitted light microscope equipped with a QImaging FAST 1394 camera and Q-Capture software. Image backgrounds were changed to black in Adobe PhotoshopCC 2015. A smoothing factor of 5 was also applied to the outline of each specimen to ensure that angular lines from pixels were rounded without altering the shape of the specimens. Morphological parameters were measured using ImageProPlus 6.2, including: area, major axis, minor axis, maximum diameter, minimum diameter, mean diameter, perimeter, roundness, length and width and derived parameters such as circularity ratio, elongation ratio, box ratio and compactness coefficient.

### 2.4 Stable isotope analyses

Single-specimen isotope analyses captures the full range of growth conditions experienced during the life-time of the foraminifera (Spindler and Dieckmann, 1986), allowing short-term variability (subannual to annual) to be assessed. However, single-specimen analysis was limited because of the low carbonate mass per individual (approximately 5ug), and low number of individuals available for analysis to define the range confidently within a single sample. Single-specimen stable isotope analysis was only carried out on samples with at least twenty Nps specimens. Conventional multi-specimen analysis was carried out on samples from sediment trap cups where there were not a sufficient number of specimens to allow for single specimen analysis, or had an abundance of specimens to allow for additional analyses. For the multi-specimen stable isotope analysis four to ten foraminiferal tests (visually inspected to be approximately 250-350 μm in size), were pooled to provide approximately 40 μg carbonate.

Prior to analysis, specimens were weighed individually on a Mettler Toledo XPR2U high precision microbalance (± 0.1 μg). All stable isotope measurements were carried out at the Department of Earth Sciences of the Vrije Universiteit in Amsterdam, The Netherlands, using a Finnigan GasBench II preparation device interfaced with a Finnigan Delta mass spectrometer. The mass spectrometer was calibrated through the international standard IAEA-CO1 and gas pressure correction was carried out using the laboratory's own in-house standard VICS. The reproducibility for the present study is based upon 54 measurements of the (external) IAEA-CO1 standard (standard mass ranging from approximately 8 to 50 ug). During the measuring period the 1 SD external reproducibility is 0.13‰ for $\delta^{13}$C and 0.11 ‰ for $\delta^{18}$O. All isotope measurements are reported as per mil deviation from the Vienna Pee Dee Belemnite scale (‰ VPDB).



## 2.5 Predicted δ18O calcite equilibrium and habitat depth

Equilibrium calcite $\delta^{18}O$ ($\delta^{18}O_{eq}$) values were calculated using Equation 1 (Peeters et al., 2002;Kim and O'Neil, 1997), to constrain Nps calcification depth at the sediment trap site, and to determine any apparent vital effects on $\delta^{18}O$ values:

$$\delta^{18}O_{eq} = 25.778 - 3.333 \times \sqrt{(43.704 + T)} + \delta^{18}O_{sw(VPDB)} + \delta^{18}O_{VE} \tag{1}$$

where T is the seawater temperature (°C) and the $\delta^{18}O_{sw}$ (on the VPDB scale) is the $\delta^{18}O$ of seawater, which we modelled and verified using published data due to the lack of year-round observations (Model results and data shown in Supplementary Information; Figure S13; (Meredith et al., 2013;Zweng et al., 2013), and $\delta^{18}O_{VE}$ is a 'vital effect' term added to correct for a potential constant offset (e.g. sensu Peeters et al., 2002)). One of our aims is to better understand the biotic impacts on $\delta^{18}O$ of planktic foraminiferal calcite so no correction for vital effects was applied ($\delta^{18}O_{VE}$ was assumed zero). V-SMOW values were converted to the V-PDB scale by subtracting 0.27 ‰ from the sea water $\delta^{18}O_{sw}$ value measured against the Vienna-Standard Mean Ocean Water (V-SMOW) scale. VPDB values were converted to the Standard Mean Ocean Water (SMOW) scale (Hut, 1987), and temperature was obtained from World Ocean Atlas 13 (WOA13; (Locarnini et al., 2013)).

## 3 Results

### 3.1 Foraminiferal flux measurements

*Neogloboquadrina pachyderma* sensu stricto (Nps) was the only foraminiferal species found in the sediment trap samples. Nps test flux generally ranged over two orders of magnitude from zero in winter months to over 300 tests m$^{-2}$ day$^{-1}$ in summer (Figure 2; Supplementary Information Table S1). Exceptionally high fluxes of 1100 - 9500 tests m$^{-2}$ day$^{-1}$ were observed during the spring of 2010 and exceptionally low summer fluxes of ~5 tests m$^{-2}$ day$^{-1}$ in 2008, although early summer fluxes that year were higher at 13-26 tests m$^{-2}$ day$^{-1}$.

Nps flux displays a double peak in some years (Figure 2; for weekly averaged data see Supplementary Information, Figure S1, calculated using von Gyldenfeldt et al. (2000)), similar to that observed in the North Atlantic by Tolderlund (1971) and Jonkers et al. (2010). In other years, only one clear peak can be identified similar to studies conducted in the northern end of the WAP, the Weddell Sea (Donner and Wefer, 1994) and in the Arctic, where a single summer peak in flux is observed (Kohfeld et al., 1996;Bauch et al., 1997;Simstich et al., 2003). Inter-annual variability is observed in the timing of peaks in Nps flux and the amplitude, which varies by three orders of magnitude during the six-year record. Inter-annual variability in Nps flux is evident from statistically-determined high outliers (defined by using interquartile ranges;



Supplementary Information, Table S1) that occur during January 2006, January 2007, October 2010, November 2010, and December 2010, March 2011, January 2012 and December 2012 and in January 2013. These higher Nps flux values mostly occur during times when carbon flux is high and/or Chl *a* concentration is above or near 3 mg m$^{-3}$, although these factors are not always associated with elevated foraminiferal flux.

### 3.2 Morphometric results

To assess the comparability of the automated and manually-derived morphometric datasets, we compared the normalised maximum diameter (MD) of specimens (Moller et al., 2013) in samples analysed by both methods (n=30; Supplementary Information Table S5; Figure S9) as this parameter is least impacted by specimen orientation (Schmidt et al.,

2003). The data are highly correlated ($R^2 > 0.8$; Figure S10), but there is a significant difference between the median (MD) of 5 of the 30 pairs of samples. Therefore, the datasets were not combined, although both methods record the same broad-scale morphological variability (Supplementary Information, Table S6).

MD, sphericity and mean grey value, measured using automated microscopy, illustrate intra- and inter-annual variability in Nps size and calcification (Figure 3). The spring-summer populations (October-April) are consistently larger than the winter (June-September) populations of the same year. While size varies seasonally, the sphericity, i.e. shape,

remains relatively unchanged (Figure 3) as expected in a mono-specific assemblage of predominantly round specimens. The translucency of the test, which we consider a proxy for test calcification, is indicated by the mean grey value (Figure 3) and shows clear intra-annual variability following MD. The timing of grey value peaks peaks (i.e., relatively thicker tests) can vary within a season, however; winter mean grey values are generally lower than the summer mean grey values in the same

year (Figure 3). We interpret this as more specimens with a thick calcite crust being collected in the summer than winter.

Statistical analysis was only carried out on morphometric data collected using the manual method (Figure 4) due to the large variability present in the secondary morphological parameters calculated from surface area and MD of the automated measurements (see Supplementary information).  Samples with the mid-date of collection period 9/12/2008, 16/05/2011, 22/02/2012 and 13/11/2013 were removed for statistical significance due to the number of specimens being

below the threshold (n=15) identified through a rarefaction analysis. The log-transformed MD data of the manual measurements had a unimodal distribution (Supplementary Information; Figure S3). The log-transformed MD dataset covering the entire time-series (2006-2013) displays both statistically significant inter-annual variability (Levene's test, p = <0.0001) and intra-annual variability (applied to 2012, p = <0.0001) in size.

Fourteen of the 32 samples display a non-normal distribution of size-invariant parameters. Principal component

analysis (PCA) of the distributions of the four size-invariant morphological parameters that relate to shape (circularity ratio, box ratio, elongation ratio and compactness coefficient) reveals two statistically defined clusters (Figure 5C). The two



clusters may represent a pre-adult and an adult life stage (Hemleben and Spindler, 1989;Vautravers et al., 2013), as
qualitative assessment shows that one cluster lacks the distinctive calcite crust associated with gametogenesis (Figure 5
A,B).

## 3.3 Oxygen and carbon isotopic composition of *Neogloboquadrina pachyderma*

A range of approximately 1‰ is observed in the single specimen analyses of Nps $\delta^{18}O$ ($\delta^{18}O_{np}$) and $\delta^{13}C$ ($\delta^{13}C_{np}$)
2006-2013 time series (Figure 6). $\delta^{18}O_{np}$ varies between +2.77‰ (17/07/2012) and +3.71‰ (11/11/2006) and has an average
value of +3.29±0.21‰ (1SD). $\delta^{13}C_{np}$ varies between +0.17‰ (22/02/2012) and +1.02‰ (02/12/2012) with an average value
of +0.56±0.20‰ (1SD). Anderson-Darling tests for normality reveal that $\delta^{18}O_{np}$ does not significantly deviate from a normal
distribution (p=0.333), while $\delta^{13}C_{np}$ values appear not normally distributed (p=0.032) (Supplementary Information, Figure
S11).

Single-specimen $\delta^{18}O_{np}$ exhibits statistically significant differences in variance between the samples (Levene's test,
p=0.005) with austral winter (May-September) samples characterised by lower $\delta^{18}O_{np}$ ratios than those collected during the
summer (December-February) months (Figure 7). There were no significant differences in variance between samples in the
single-specimen $\delta^{13}C_{np}$ dataset (Levene's test, p=0.076).

## 4 Discussion

### 4.1 Stable isotopes and depth ranges of *Neogloboquadrina pachyderma*

The range of both $\delta^{13}C_{np}$ and $\delta^{18}O_{np}$ values varied between the bulk and single Nps specimen analysis (Figure 6).
The multi-specimen data displays only a 1‰ range in both isotope ratios across the record, whereas a 4‰ and a 2‰ range is
recorded in the single-specimen $\delta^{18}O_{np}$ and $\delta^{13}C_{np}$ samples, respectively.  Comparison of the mean single-specimen $\delta^{18}O_{np}$
and the multi-specimen $\delta^{18}O_{np}$ (Figure 6) from the same cup shows that the average values of single-specimen $\delta^{18}O_{np}$ are in
most – but not all – cases lower than multi-specimen $\delta^{18}O_{np}$. The source of this offset likely stems from how the single-
specimen averages are calculated. For multi-specimen analysis, the contribution of each foraminiferal test to the final sample
$\delta^{18}O_{np}$ is proportional to its mass. As higher mass/larger Nps tests tend to be covered by a secondary crust precipitated at
depth (Kohfeld et al., 1996) with higher $\delta^{18}O_{np}$ (Bauch et al., 1997), the resulting multi-specimen average $\delta^{18}O_{np}$ tends to be
slightly higher than the mean of the single-specimen $\delta^{18}O_{np}$.





Assuming no vital effect, the predicted $\delta^{18}O_{eq}$ at 50-100 m water depth show similar patterns to the measured $\delta^{18}O_{np}$ in spring to early autumn, but diverges in late autumn and into winter (Figure 6). This observation suggests that the summer Nps calcification depth lies between 50 and 100 m and likely becomes shallower in winter (Figure S13). This observation agrees with previous assessments in both northern and southern polar waters (Hendry et al., 2009;Kohfeld et al., 1996;Bauch et al., 1997). However, there remains considerable uncertainty in the extent of vital effects in Nps oxygen isotope fractionation, with studies reporting both calcification in equilibrium (King and Howard, 2005;Jonkers et al., 2013) and out of equilibrium with seawater (Kohfeld et al., 1996;Bauch et al., 1997;Simstich et al., 2003;King and Howard, 2005;Mortyn and Charles, 2003), the latter of which can depend upon ontogenetic stage with the secondary crust characterised by relatively high $\delta^{18}O$ values (Kozdon et al., 2009).

## 4.2 What controls the export flux of foraminifera?

A qualitative view of our flux data reveals pronounced interannual variability, indicating that there is no direct control of foraminiferal flux specifically due to seasonal changes in water column conditions. Spearman's Rank analysis indicates that there are significant correlations for Nps flux only with organic carbon and organic nitrogen flux (Supplementary Information, Table S8, Figure S10), with highest fluxes between November and February associated with phytoplankton blooms. Both organic carbon and nitrogen fluxes correlate with other environmental parameters, such as SST, Chl $a$, and SIC. The lack of significant correlation between Nps flux and environmental variables means that one parameter alone cannot explain the flux pattern. While temperature is commonly the dominant control on foraminiferal distribution, the range of temperatures experienced at the sediment trap site throughout the year, from +2°C to -1°C, is limited compared to the full temperature range of the species (Schmidt et al., 2006). Rather, it is the effects of environmental variables on food availability that determines Nps flux, as the combination of SST, Chl $a$ and sea ice determines the strength of primary productivity. This relationship between foraminiferal flux and primary production is similar to that observed in other high-latitude regions (Donner and Wefer, 1994;Kohfeld et al., 1996;Kuroyanagi et al., 2011) and, indeed, globally (Tolderlund, 1971). Sediment trap data in the North Pacific suggests that Nps fluxes are highest when the water column is well mixed, and phosphate, silicate and nitrate are highest (Reynolds and Thunell, 1986;Jonkers and Kučera, 2015).

## 4.3 What controls morphology and geochemistry?

### 4.3.1 Relationship between *Neogloboquadrina pachyderma* size and single-specimen stable isotopes

*Neogloboquadrina pachyderma* have been shown both to exhibit a co-variation and lack of correlation between test size and stable isotope values, likely as a result of regional differences in environmental controls on the two parameters (Bauch et al., 1997;Jonkers et al., 2013;Kuroyanagi et al., 2011;Peeters et al., 2002;Ezard et al., 2015). There is a consistent size effect on both the $\delta^{18}O_{np}$ and $\delta^{13}C_{np}$ across all our data ($\delta^{18}O_{np}$ r = 0.52; $\delta^{13}C_{np}$ r = 0.23, n = 191) which is maintained for



$\delta^{18}O_{np}$ when divided into the 150-250 µm (r = 0.28, n = 89) and >250µm (r = 0.32, n = 102) fractions. In comparison, there is no apparent size effect on $\delta^{13}C_{np}$ (150-250 µm r = 0.007, n = 89; >250 µm r = 0.06, n = 102) as expected for a non-symbiont bearing species (Figure S14). Therefore, we can rule out size-specific kinetic/metabolic effects on $\delta^{18}O_{np}$ values and suggest that size-specific differences are largely related to variable oceanographic conditions, life stage, and variability in calcification depth (discussed in Section 4.4).

### 4.3.2 The role of environmental variability on *Neogloboquadrina pachyderma* test size and morphology

PCA revealed that seasonality alone cannot explain all of the observed morphological variability (Supplementary Information). Redundancy analysis (RDA) of the means shows a single dominant trend in the joint space of the manually collected size-normalised, size-dependent, size-invariant morphological data (see Supplementary Information for definitions) and the environmental parameters (Figure 8). The first canonical axis (F1, 84.9% of the joint variation) is dominated by elongation and circularity, and is negatively linked to SIC. As the dominant food of *N. pachyderma* is diatoms (Spindler and Dieckmann, 1986), we suggest that the presence of sea ice has an adverse effect on the population by limiting habitat size and food availability; smaller and less round specimens without a calcite crust suggest reduced frequency of gametogenesis.

The second canonical axis (F2, 10% of the joint variance) shows the opposite effects of Chl *a* and SST on compactness, which in turn indicates the presence of the two growth stages (Figure 5A and B). A negative correlation is observed between the compactness and SST, nitrogen and carbon fluxes, and a positive correlation is observed between Chl *a* concentration and compactness (Figure 8). These trends show the impact of primary productivity changes on the abundance of the two morphologies, where higher Chl *a* concentration is associated with more compact tests indicative of reproductive success and post-gametogenesis secondary calcification (Kohfeld et al., 1996;Eynaud et al., 2009). Nutrient concentrations and phytoplankton availability has also been observed to influence Nps test shape and size in the North Atlantic and Arctic Oceans (Eynaud et al., 2009;Moller et al., 2013). Additionally, the constrained and unconstrained variances of the RDA indicate that the environmental parameters observed at the sediment trap site between 2006 and 2013 are only responsible for 50% of the morphological variability indicating that other factors – potentially related to foraminiferal physiology or non-cryptic genetic variability – are additionally important.

### 4.4 A typical year at the Palmer LTER sediment trap site

To summarise the ecological drivers on Nps flux, size and morphology, we describe a composite year (Supplementary Information, Figure S1, S2) divided into six distinct phases (Figure 9) based on the intra-annual trends of Nps flux and the environmental variables.

*Phase 1 – Winter sea-ice*



During the Antarctic winter, Nps dwell at shallow depths, just below or within the sea ice (Hendry et al., 2009). The lowest Nps fluxes and smallest test sizes of the year occur during the onset of winter sea ice and peak sea ice concentration (SIC). Sea ice occurs between July and October, the coldest months, and, on average, SIC peaks at the end of winter, during September, although there is some inter-annual variability. By the end of winter, seawater at 100 m water depth is warmer

than at the surface due to the upward mixing of heat from modified upper CDW, the core of which is present deeper than the sediment trap depth (Smith and Klinck, 2002). At the same time, salinity at depth (100 m) is increased due to brine rejection during sea ice formation, which also contributes to the deepening of the mixed layer and the erosion of stratification. Exposure to this seasonal increase in salinity is not expected to result in mortality because Nps are known to tolerate very high salinities in ice brines (Spindler and Dieckmann, 1986). However, sea ice presence also coincides with the lowest

organic carbon and nitrogen fluxes, suggesting lower food availability for the foraminifera limiting growth and energy for reproduction. Specimens overwintering under these conditions appear to enter a hibernation-like state where they cease growing and do not reproduce. The lack of secondary encrustation results in more transparent tests with low $\delta^{18}O_{np}$.

*Phase 2 - Sea ice break up*

Spring sea ice break-up and melt results in decreased surface salinity. The onset of shallow stratification of the water column and release of nutrients from sea ice and glacial melt provides an ideal setting for diatom blooms. As a result, Chl *a* concentration steadily rises during this phase together with an increase in organic carbon flux. The increased food availability results in an increased Nps flux, growth to larger sizes and increases in the number of specimens which are going through their full reproductive cycle (based on test grey values). All of this evidence suggests environmental conditions

become more favourable for the species, triggering a completion of their life cycle and the start of migration within the water column. Whilst melting sea ice has little impact on $\delta^{18}O_{np}$, surface glacial meltwater and precipitation results in low $\delta^{18}O_{sw}$ at this time of year, which would result in lower $\delta^{18}O_{np}$ in surface dwelling specimens. However, by the beginning of summer as Nps migrate deeper and undergo gametogenesis, $\delta^{18}O_{np}$ trends towards higher values due to lower ambient temperatures and a greater proportion of secondary calcite crust formation (Kozdon et al., 2009).

*Phase 3 – Summer*

Highest Nps fluxes at the sediment trap site are associated with the warmest time of the year and the complete disappearance of sea ice by the end of November. Between late November and mid-January SSTs continue to increase, and surface waters freshen, resulting in stronger surface stratification than during sea ice break up (Phase 2). The melting sea ice

and the development of shallow stratification leads to increased food availability, which is reflected in the Chl *a* concentration. Chl *a* concentration reaches its peak during this phase and remains stable and high during the summer months, organic carbon flux also increases. Inter-annual variability can be seen in both the Chl *a* concentration and organic carbon flux records during this phase.





In late spring-early summer, Nps flux peaks. The majority of the Nps complete their full life cycle indicated by their large spherical tests with secondary calcite crust. On average, $\delta^{18}O_{np}$ shows no significant offset from $\delta^{18}O_{eq}$ within the well mixed surface layer (Figure S2). Small, relatively elongated, non-encrusted specimens are also captured by the sediment trap during late spring-early summer, albeit in small numbers. These small specimens are more transparent, have lower $\delta^{18}O_{np}$,
and are likely immature specimens that have not reproduced prior to mortality. Export of juvenile specimens, for example by storms, is a common mechanism that interrupts the foraminiferal life cycle (Schiebel et al., 1995).

*Phase 4 - Late summer*

During the late summer phase, Nps flux decreases despite surface warming and increasing organic carbon flux,
which peaks by mid-February. By this time, surface water stratification reaches its maximum and salinities their minimum due to input of glacial meltwater from the coastal region. Chl *a* concentration remains relatively high. $\delta^{18}O_{np}$ is at its highest during the summer months, despite low seawater oxygen isotope composition throughout the habitat range of the foraminifera. This high $\delta^{18}O_{np}$ reflects the largest test sizes with the highest proportion of secondary encrustation in specimens that have been through their whole life cycle. However, relatively small and translucent tests, with low $\delta^{18}O_{np}$, are
also recorded in the summer months of some years (e.g. 2012), which could be interpreted as a reduction in gametogenesis due to less favourable growth conditions than those experienced by specimens that calcified during other years.

*Phase 5 - Autumn*

By the end of summer, the Nps depth range contracts with foraminifera living closer to the sea surface. In the
autumn Nps flux increases again while carbon flux and SST decrease. In response to cooling away from optimum temperatures, test sizes decline and fewer specimens reach reproductive maturity. During the second half of February, surface stratification decreases and subsurface water freshens and warms due to mixing with surface waters. The early part of this phase is characterised by a second peak in Chl *a* concentration, accompanied by high organic carbon flux, perhaps related to the end of season flux of diatom resting spores out of the surface water (e.g. Pike et al., 2009). By the end of April
there is no new primary production resulting in a gradual decline of organic carbon flux. Samples from the autumn months have low $\delta^{18}O_{np}$ due to the lack of secondary encrustation, and subsurface freshening and warming (Figure S2).

*Phase 6 – Early winter*

Nps fluxes are very low during early winter in response to decreasing organic carbon flux close to zero by May and
sea water temperature cooling to freezing. The cooling leads to the erosion of surface water stratification and to deepening of the mixed layer. A decline in food availability results in smaller foraminiferal sizes, together with enhanced mortality rates during periods of sea ice formation (Vautravers et al., 2013).





$\delta^{18}O_{np}$ shows striking inter-annual variability during the winter, and the largest mismatch of the year with $\delta^{18}O_{eq}$. We suggest that Nps become dormant in sea ice during the winter phase and, hence, $\delta^{18}O_{np}$ of these overwintering specimens still reflects the conditions of the preceding autumn or summer.

## 4.5 Interannual variability and extreme events: the role of the Southern Annular Mode and the El Niño-Southern Oscillation

Anomalously large numbers of Nps (approximately 9500 individuals per day, compared to an overall mean flux of 300 individuals per day) sank into the sediment trap during October and November 2010 (Figure 2). These fluxes are unprecedented in the six-year time series and occurred at an earlier time of the year than flux peaks in other years. The specimens were some of the largest and roundest recorded with high $\delta^{18}O_{np}$ values, and a high proportion having a secondary crust. The flux, morphology and stable isotope records of this 2010 event indicate that the environmental conditions were optimal for Nps growth and reproduction. However, the reasons behind this early flux event are not immediately clear from the observed environmental parameters.

Nps flux, morphology and stable isotope composition are all closely linked to sea-ice extent and food availability. Our records show that differences in the timing and amplitude of peak Nps flux between 2006 and 2012 are driven by the timing of the onset of sea-ice melt. Additionally, during periods of extensive sea ice cover, smaller specimens are more abundant. In contrast, periods of high food availability (spring-summer period and/or lower sea ice concentration) and reproductive success result in higher $\delta^{18}O_{np}$. Based on these findings, the most likely explanation for the 2010 flux event is the combination of early sea-ice retreat and/or low SIC and an early increase in primary production, stimulating foraminiferal growth and reproduction.

Sea ice has a complex but important relationship with the El Niño Southern Oscillation (ENSO) and the Southern Annular Mode (SAM) (Turner, 2004) and some of the inter-annual variability in Nps flux, morphometric and stable isotope data may be driven by the impact of the ENSO and SAM on WAP oceanographic conditions. Increasingly positive SAM and a predominance of strong La Niña events since the 1990s (Figure 10) resulted in the development of strong negative sea level pressure (SLP) anomalies along the WAP. This supported anomalously warm northerly winds in the Bellingshausen Sea, and was associated with earlier wind-driven sea-ice retreat and later sea-ice advance. Conversely, positive SLP anomalies during the autumn months in the Amundsen Sea region are associated with cold southerly winds over the WAP region resulting in early sea-ice advance (Stammerjohn et al., 2008). In 2010, a very strong, positive SAM coincided with strong La Niña-like conditions (Figure 10; Meredith et al. (2017)). Low sea-ice extent and short sea ice duration during the winter of 2010 resulted in low sea-ice melt in the sediment trap region in the austral spring-summer season of 2010-2011 (Meredith et al., 2017), creating optimal conditions for very high flux of large, isotopically high, Nps specimens.

In comparison to 2010, 2012 was characterised by El Niño-like conditions over the Pacific Ocean, and SAM switched to a negative mode after September 2012 (Marshall and Thompson, 2016). Sea ice lingered in the region around the





sediment trap site until late November 2012 (Meredith et al., 2017). Nps flux between September and December 2012 was much lower than in 2010, and specimens that calcified during this period were, on average, smaller (Figure 10). Nps in 2012 exhibited more variable $\delta^{18}O_{np}$ values, but similar average degrees of calcification to 2010 indicating that those specimens that grew still succeeded in completing their life cycles despite the extended sea ice period. This reproductive success was

most likely due to the high Chl $a$ concentrations sustained by changes in wind forcing.

As anthropogenic forcings persist, it is expected that northerly winds will become more persistent and stronger in the future in response to a dominant positive SAM (van Wessem et al., 2015), creating the possibility of shorter sea-ice seasons and warmer SST in the WAP region. These interactions between atmospheric oscillations, sea ice and biological production exhibit complex regional behaviour and can be localised, even between different locations along the WAP (Kim

et al., 2018). Overall, these processes will create more favourable growing conditions for Nps along the WAP, therefore we speculate that their abundance during the spring-summer period will increase with greater reproductive success, increasing carbonate production along the WAP whilst carbonate saturation remains high enough to support thick test growth.

## 5 Synthesis and Outlook

At the Palmer LTER sediment trap site, Nps flux displays a large peak during the late spring-early summer period once the sea ice has completely retreated and Chl $a$ concentration is increasing. During this time, Nps appear to complete their full life cycle, based on their large spherical tests with secondary calcite crust indicating gametogenesis. As the phytoplankton bloom and Chl $a$ concentration increases and Nps can sustain their energetic needs at deeper ocean depths, their flux decreases but remains relatively high from the surface waters during the summer, generally peaking again in late

summer. During the austral autumn and into winter, as Chl $a$ concentrations and SSTs decrease, and stratification begins to break down, Nps flux decreases and fewer specimens complete their life cycle. However, because Nps remain present in the surface waters (shown by its continuous presence in the trap) some specimens must be able to survive the unfavourable conditions. Nps can survive and live within the brine channels of the sea ice (Spindler, 1996;Hendry et al., 2009), potentially surviving the winter months dormant – without calcifying – until conditions become suitable for growth during early spring

when sea ice begins to break up and melt. Interannual variability in foraminiferal flux and shape is linked to sea ice dynamics and food availability. Annual phases of Nps flux, morphology and stable isotope variability are modulated by extreme SAM and ENSO conditions, as seen during austral spring 2010.

Based on our improved understanding of Nps ecology we suggest that non-encrusted Nps specimens should not be combined with encrusted specimens for geochemical proxy analysis as the two different morphologies record different depth

habitats and seasons. Nps proxy records that only utilise encrusted specimens will likely only reconstruct austral spring and summer conditions, and may be biased towards heavier $\delta^{18}O$ values due to secondary crust formation at depth. Conversely, smaller, non-encrusted specimens may bias towards autumn and winter conditions. Investigation of the two growth stages





separately could be useful for investigating seasonality. Finally, average Nps flux during a strong positive SAM/La Niña year is much higher than average flux during a year that is not impacted by strong positive SAM and ENSO conditions, thereby overwhelming typical years in an averaged bioturbated sedimentary record and skewing palaeo-reconstructions based on foraminiferal stable isotopes.

**Data availability.** Morphometric and stable isotope data presented in this study are available from https://doi.pangaea.de/xx.xxxx/PANGAEA.xxxxxxx.

**Acknowledgements.** The authors would like to thank all the officers and crews of the US ARSV L.M. Gould and all those who have been involved in recovering and deploying the Palmer LTER sediment trap throughout the years. AM was
supported by a Cardiff University President's Scholarship, and the stable isotope analysis was funded by an Antarctic Science Ltd. International Bursary awarded to AM. The sediment trap time series has been funded by a series of awards from the US NSF Office of Polar Programs, including award PLR-1440435 to HD. KH is funded by a Royal Society University Research Fellowship and DNS is supported by a Royal Society Wolfson Merit Award. KME was supported by a Leverhulme Trust Early Career Fellowship.

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





**Figure captions:**

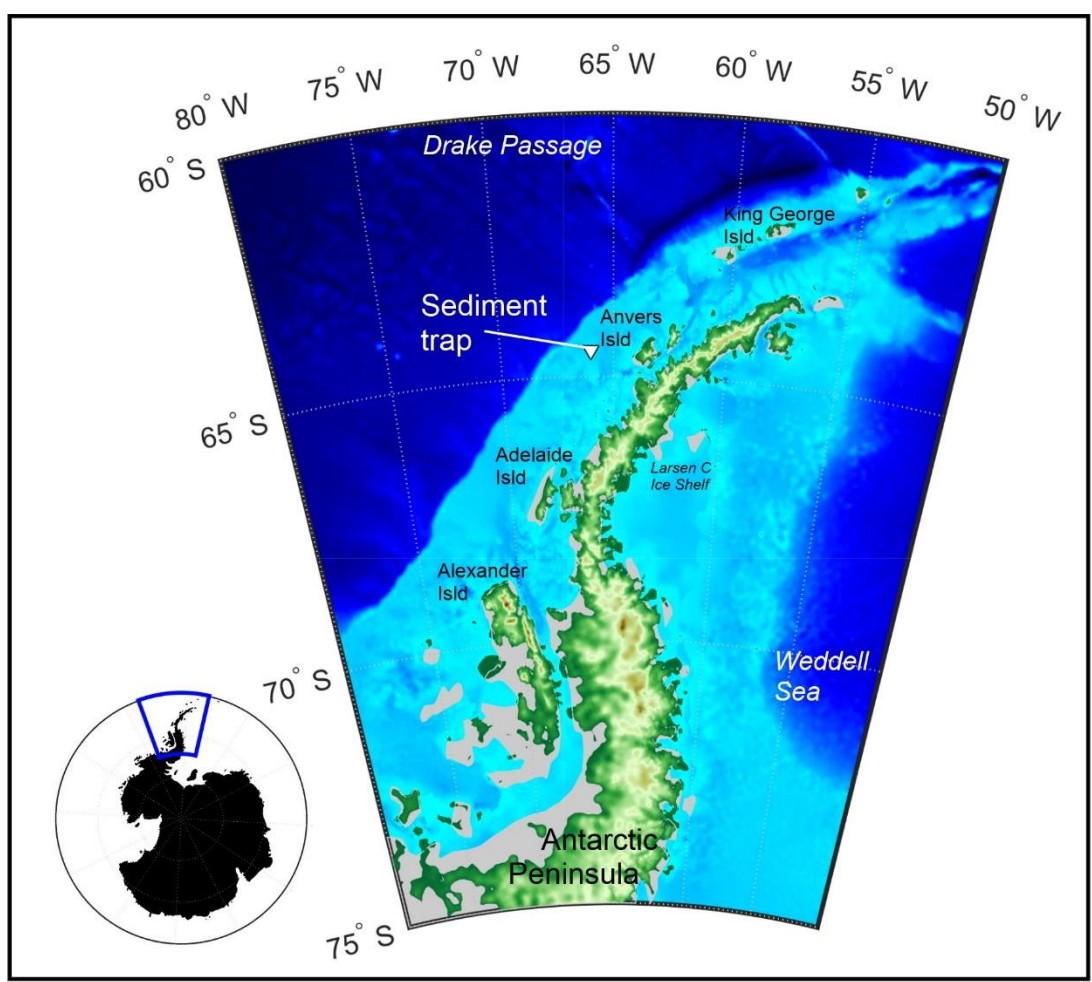

**Figure 1: A map of the study area with Palmer LTER sediment trap location marked with the triangle. Grey regions show ice shelf regions. Main map is enlarged view of blue box, and was made using etopo1 bathymetry.**





**Figure 2: Environmental parameters plotted with foraminiferal and organic matter time-series flux record from Palmer LTER sediment trap. A)** Satellite-derived weekly sea-surface temperature (grey circles corresponding to mid-point of trap cup opening times); **B)** Sea-ice concentration (SIC) at the LTER Palmer sediment trap site (purple) **C)** Chlorophyll a (Chl a) records from 5 offshore Palmer Station; **D)** Neogloboquandrina pachyderma sensu stricto (Nps) shell flux (blue diamonds); **E)** Sediment trap organic carbon flux records (green squares). AJO: April (autumn), July (winter), October (spring).



**Figure 3: Time series record of the foraminiferal morphological parameters collected by the automated analysis. A) Mean grey values as a measure of translucency and calcification; B) Sphericity; C) Maximum diameter. AJO: April (autumn), July (winter), October (spring).**







**Figure 4: Boxplot of (A) Nps surface area and (B) maximum diameter records collected by manual morphological analysis. Crosses represent outliers defined by interquartile range. Dashed lines are visual divisions between the years. The width of each box represents the length of time that the collection cup was open in the sediment trap; thinnest box is 7 days, widest box is 92 days.**





**Figure 5: Scanning electron microscope images of A) post-reproduction adult Neogloboquadrina pachyderma (with secondary crust) and B) pre-adult morphotype (showing evidence of dissolution). C) Example PCA biplot of 16/09/2007 normalised size-invariant morphological variables showing clustering of data points (blue circles). F1 is the first principal component axis; F2 is the second principal component axis. Proportion of variance explained by each axis shown in parentheses (further details are found in the Supplementary information section 3).**



**Figure 6: Multi-specimen and single-specimen adult Neogloboquadrina pachyderma (Nps) stable isotope dataset (VPDB) showing (A) $\delta^{18}O_{np}$ (blue) and (B) $\delta^{13}C_{np}$ (red). Grey horizonal bar shows the range of calculated $\delta^{18}O_{eq}$ values for the depth range 50-100m, using seawater temperature and salinity profiles in the World Ocean Atlas 13 dataset (see Supplementary Information Figure S13 for comparison of calculations to measured $\delta^{18}O_{sw}$ profiles). Error bars show ±1SD. AJO: April (autumn), July (winter), October (spring).**



**Figure 7: Boxplot of single-specimen (A) $\delta^{18}O_{np}$ and (B) $\delta^{13}C_{np}$ (VPDB). Crosses represent outliers defined by interquartile range (horizontal line represents median). Dashed lines are visual divisions between the years. The width of each box represents the length of time that the collection cup was open in the sediment trap; thinnest box is 7 days, widest box is 92 days.**





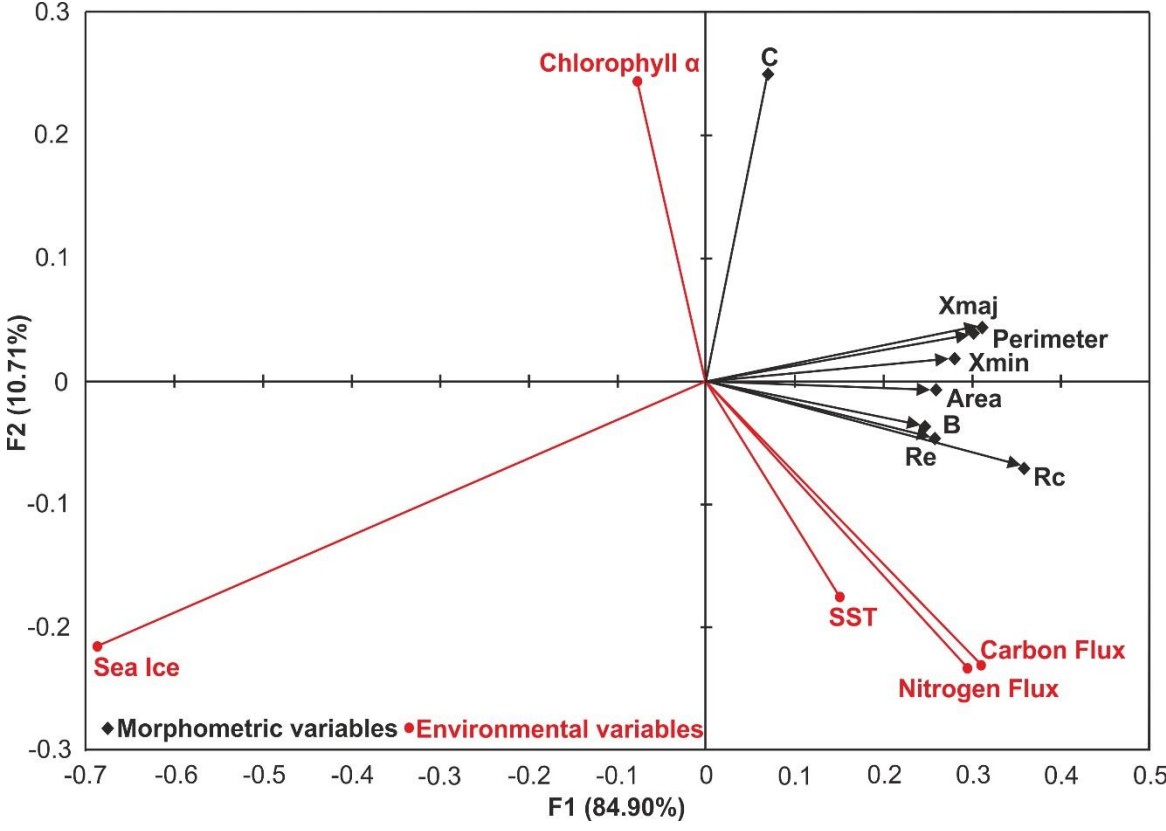

**Figure 8: Redundancy Analysis biplot of the means of the normalised size-dependent and size-invariant morphological data (black diamond) and the environmental parameters (red circle). Xmaj = maximum diameter, Xmin = minimum diameter, B = Box ratio, Rc = Circularity ratio, Re = Elongation ratio, C = Compactness coefficient. See supplementary information Table S2 for derivation of secondary morphological parameters.**





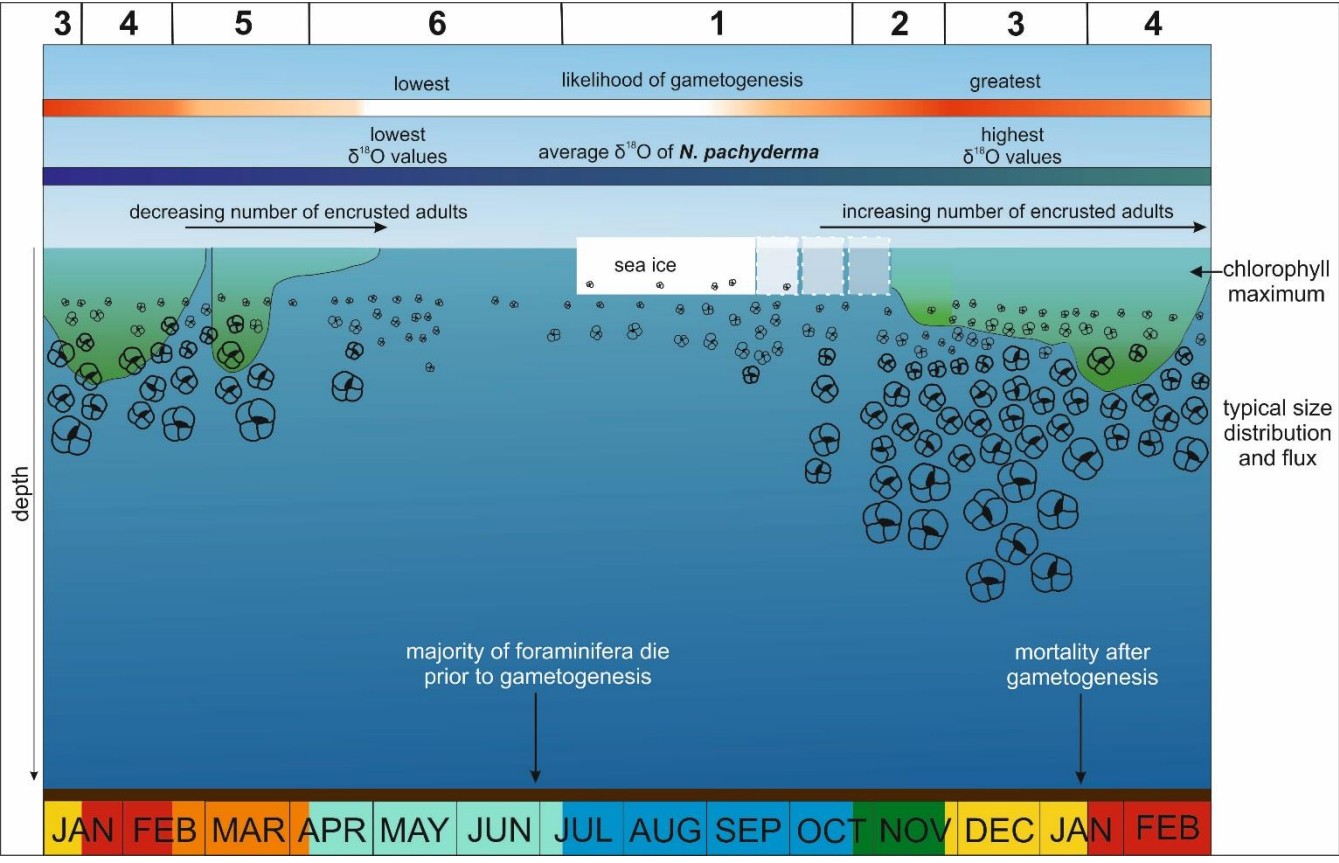

**Figure 9: Summary schematic of a typical year at the Palmer sediment trap site. Colours illustrate the six phases of the annual cycle described in the main text.**



**Figure 10: (A) Multi-specimen (purple) and single-specimen (pink) mean d18Onp (error bars show ±1SD). (B) Total Nps flux (black diamonds show statistically significant outliers, defined by interquartile range). (C) Southern Annular Mode (SAM) index and (D) El Niño Southern Oscillation (ENSO) index. SAM index is based on the departure from the zonal means between 40°S and**
5  **65°S (Marshall, 2003). Red sections show positive SAM, blue sections show negative SAM. The ENSO index is based on the departure from the 1950-1993 reference period (Wolter, 2016). Red sections show El Niño-like conditions, blue sections show La Niña-like conditions. Note the co-variation between positive SAM and La Niña-like conditions, negative SAM and El Niño-like conditions, and the strong positive SAM, La Niña, and high Nps flux in 2010.**