# Peer review of "Temporal variability in foraminiferal morphology and geochemistry at the West Antarctic Peninsula: a sediment trap study"

_Biogeosciences, 2019_

## Short Comment (SC1) · 19 Feb 2019

The statements about converting V-SMOW to V-PDB on Page 6, lines 5-15 are not correct. The -0.27‰ offset does not convert SMOW to PDB - this is a an offset correction that is necessary because we use isotope values from two different standardizations in this empirical equation. Note that the SMOW and PDB scales are ∼30‰ different so -0.27‰ does not convert one to the other as you state. Please reword this in your methods so that the statement is accurate.

---

## Referee Comment (RC1) · Gerald M. Ganssen (Referee) · 21 Feb 2019

The paper by Mikis et al. provides a six year long sediment trap time series showing the response of the planktonic foraminifer Neogloboquadrina pachyderma. It presents data on shell flux, morphology and stable isotopic composition of this species in the extreme environment next to Antarctica and relates these data to changes of measured environmental parameters. The paper is well-organized, well-written and hence guides the reader nicely through the story. Methods are clearly explained, statistics performed are appropriate, and by clearly describing the species/morphotype (good photographs) it prevents any misunderstanding compared to earlier papers on this species. Results

and interpretations are substantiated by providing additional information in the supplementary information. Results are clearly presented and discussions are convincing supported by the figures and tables. The set-up of the discussion section is elegant by putting questions mirroring the objectives of the research and giving answers to these questions by own data and putting the findings into the context of other research in this field. It shows the importance of seasonal and inter-annual variability influencing the measured parameters and showing the complexity. These findings should be taken into account when working on sediment material. It further provides an outlook how this ecosystem might change as the result of current and future climate changes. Figure 9 nicely summarizes the most important findings showing a typical annual cycle.

In the following I address a few point of attention to be changed: p.3, l.14: delete bracket after "site" p.6,l.11-16: only mention the conversion from VSMOW to VPDB as you show all data against the VPDB scale. In addition to the reference of Hut, 1987, which is difficult to get, use Coplen, 1988, Chemical Geology, 71, p293-297, which explains the normalization of isotope data.

p.9, l.1: add data in between "d18Oeq" and "at"

p.13, l.19: the paper would be improved if satellite images could be given to document this extreme flux from 2010, showing the sea-ice retreat and/or increase in primary production

p.21,fig.4: enlarge the font of the vertical axis numbers in the upper part of the figure

p.22, figure caption: species name in italics, same for fig.6 caption and maybe others

p.27, on the axis in the two lowermost figures: either Departure or departure

---

## Referee Comment (RC2) · 17 Jul 2019

sepulcre sophie (Referee)

sophie.sepulcre@u-psud.fr

The manuscript "Temporal variability in foraminiferal morphology and geochemistry at the West Antarctic Peninsula: a sediment trap study" by Mikis et al. provides an original work about the polar planktonic foraminifera Neogloboquadrina pachyderma in the West Antarctic Peninsula. This study consists of a six-year long sediment trap study, and provides data about the shell fluxes, morphology, and stable isotope variability of N. pachyderma. These results allow to document the ecology of this species, for which only a few data exist, and to discuss the influence of various environmental parameters on the observed inter and intra annual variability in the N. pachyderma records. A lot of

results are exhibited in the manuscript, and several statistical tools are used to reinforce the interpretation. Morever, a lot of additional figures and Tables help to understand the discussion. The paper is well-written and easy to read and to understand, with a very nice synthesis given by Figure 9 and explained in Section 4.4. I would suggest to clarify some points in order to improve the quality of the paper, as suggested below:

- p9 l1: Section 4.1: I do not understand why the authors assume that there is no vital effect whereas they have all the data to discuss it;

- p9 l1: Section 4.2: In the first sentence, the authors wrote: "there is no direct control of foraminiferal flux specifically due to seasonal changes in water column conditions" whereas in the Results section 3.1 they describe that "Nps test flux generally ranged over two orders of magnitude from zero in winter months to over 300 tests m-2 day-1 in summer", could you clarify this point?;

- p10 l4-6: Section 4.3.1: I do not understand why the authors discarded the size-specific kinetic/metabolic effects on $\delta18O_{np}$ ;

- Section 4.4 : maybe the authors could add some informations about the seasonal cycle of the diatoms production in the area, is there any literature about that?

As more general comments:

- I think that the $\delta13C$ record could be better interpretetd/used: is there any relation-ships with the chlorophyll maximum ? With the nutrient proxies ? The primary productivity ?

- You should discuss the potential impact of the carbonate ion concentration on the shell thickness as well as on the $\delta18O$;

- What consequences the results of this work can have for paleoclimate studies ? To finish, a very small error appears p7 l18: "peaks" written 2 times.

Despite these few comments/suggestions, I think this paper is a very nice contribution to the better understanding of the carbonate productivity in this area, and to the ecological constrains on N. pachyderma.

---

## Author Comment (AC1) · 31 Jul 2019

Many thanks for the helpful comment. This section has now been reworded accordingly:

"V-SMOW values corrected for the offset generated by the use of two different standardizations in Equation 1 by subtracting 0.27 ‰ from the sea water $\delta$ 18Osw value measured against the Vienna-Standard Mean Ocean Water (V-SMOW) scale (Hut, 1987;Coplen, 1988), and temperature was obtained from World Ocean Atlas 13 (WOA13; (Locarnini et al., 2013))."

---

## Author Response (AR1)

[revised manuscript text omitted]

**Referee one (Ganssen)**

The paper by Mikis et al. provides a six year long sediment trap time series showing the response of the planktonic foraminifer Neogloboquadrina pachyderma. It presents data on shell flux, morphology and stable isotopic composition of this species in the extreme environment next to Antarctica and relates these data to changes of measured environmental parameters. The paper is

5   well-organized, well-written and hence guides the reader nicely through the story. Methods are clearly explained, statistics performed are appropriate, and by clearly describing the species/morphotype (good photographs) it prevents any misunderstanding compared to earlier papers on this species. Results and interpretations are substantiated by providing additional information in the supplementary information. Results are clearly presented and discussions are convincing supported by the figures and tables. The set-up of the discussion section is elegant by putting questions mirroring the objectives of the

10  research and giving answers to these questions by own data and putting the findings into the context of other research in this field. It shows the importance of seasonal and inter-annual variability influencing the measured parameters and showing the complexity. These findings should be taken into account when working on sediment material. It further provides an outlook how this ecosystem might change as the result of current and future climate changes. Figure 9 nicely summarizes the most important findings showing a typical annual cycle.

15  *We thank the reviewer for the positive comments, and excellent suggestions for how to improve the manuscript. We have responded to the points raised, highlighted in bold and italics below.*

In the following I address a few point of attention to be changed:

p.3, l.14: delete bracket after "site"

*Done*

20  p.6,l.11-16: only mention the conversion from VSMOW to VPDB as you show all data against the VPDB scale. In addition to the reference of Hut, 1987, which is difficult to get, use Coplen, 1988, Chemical Geology, 71, p293-297, which explains the normalization of isotope data.

*Done, and reference added. Please note that we have also corrected the text on page 6, as commented on by H. Spero.*

p.9, l.1: add data in between "d18Oeq" and "at"

25  *We have changed this sentence to: "Assuming no vital effect, the predicted $\delta^{18}O_{eq}$ values at 50-100 m water depth show similar patterns to the measured $\delta^{18}O_{np}$ in spring to early autumn, but diverges in late autumn and into winter".*

p.13, l.19: the paper would be improved if satellite images could be given to document this extreme flux from 2010, showing the sea-ice retreat and/or increase in primary production

30  *Visuals of the anomalous sea-ice conditions are found in Figure 9 of Meredith et al., 2017. This is now included in the text, to direct the reader. We have reworded this section for clarity, and added in a supplementary figure to illustrate (Figure S15):*

*"Early ice retreat following average sea-ice conditions during the winter of 2010 (Figure S15, and see Figure 9 in Meredith et al., 2017), resulted in optimal conditions for very high flux of large, isotopically high, Nps specimens in the sediment trap region in the austral spring-summer season of 2010-2011."*

p.21,fig.4: enlarge the font of the vertical axis numbers in the upper part of the figure

35  *Done*

p.22, figure caption: species name in italics, same for fig.6 caption and maybe others

*__Done__*

p.27, on the axis in the two lowermost figures: either Departure or departure

*__Done__*

**Referee two (Sepulcre)**

The manuscript "Temporal variability in foraminiferal morphology and geochemistry at the West Antarctic Peninsula: a sediment trap study" by Mikis et al. provides an original work about the polar planktonic foraminifera Neogloboquadrina pachyderma in the

10 West Antarctic Peninsula. This study consists of a six-year long sediment trap study, and provides data about the shell fluxes, morphology, and stable isotope variability of N. pachyderma. These results allow to document the ecology of this species, for which only a few data exist, and to discuss the influence of various environmental parameters on the observed inter and intra annual variability in the N. pachyderma records. A lot of results are exhibited in the manuscript, and several statistical tools are used to reinforce the interpretation. Moreover, a lot of additional figures and Tables help to understand the discussion. The paper is well-

15 written and easy to read and to understand, with a very nice synthesis given by Figure 9 and explained in Section 4.4. I would suggest to clarify some points in order to improve the quality of the paper, as suggested below:

*__We thank the reviewer for the positive comments, and excellent suggestions for how to improve the manuscript. We have responded to the points raised, highlighted in bold and italics below.__*

- p9 l1: Section 4.1: I do not understand why the authors assume that there is no vital effect whereas they have all the data to discuss
20 it;

*__As we do not have the co-located water samples from where the foraminifera were growing, nor can we constrain the exact depth of calcification, we took the decision that it was not possible to measure the vital effect and, so, made the simplest assumption that this vital effect was negligible. This assumption allows the reconstruction of reasonable depths of calcification using seawater $\delta^{18}O$ profiles.__*

25 - p9 l1: Section 4.2: In the first sentence, the authors wrote: "there is no direct control of foraminiferal flux specifically due to seasonal changes in water column conditions" whereas in the Results section 3.1 they describe that "Nps test flux generally ranged over two orders of magnitude from zero in winter months to over 300 tests m-2 day-1 in summer", could you clarify this point?;

*Apologies if this was not clear. Our point was that whilst there are seasonal changes in flux, there is also strong interannual variability, so that there is no simple relationship between season and foraminiferal flux. We have changed the first statement to:*

*"A qualitative view of our flux data reveals that, whilst there are generally fewer foraminifera in winter than summer, there is also pronounced interannual variability, indicating that there are complex controls on foraminiferal flux in addition to seasonal climatologies of water column conditions."*

- p10 l4-6: Section 4.3.1: I do not understand why the authors discarded the size- specific kinetic/metabolic effects on δ18Onp ;

*We have added to this discussion for clarification:*

*"There is a consistent size effect on both the $\delta^{18}O_{np}$ and $\delta^{13}C_{np}$ across all our data ($\delta^{18}O_{np}$ r = 0.52; $\delta^{13}C_{np}$ r = 0.23, n = 191) which is only weakly maintained for $\delta^{18}O_{np}$ when divided into the 150-250 µm (r = 0.28, n = 89) and >250µm (r = 0.32, n = 102) fractions. In addition, there is no significant offset between the two size fractions with mean $\delta^{18}O_{np}$ values of +2.72 ± 0.59 and +3.21 ± 0.34 ‰ for the 150-250 µm and .>250µm size fractions respectively (1SD)."*

- Section 4.4 : maybe the authors could add some informations about the seasonal cycle of the diatoms production in the area, is there any literature about that?

*Given that we have Chl a data, not diatom data, we have decided not to include too much additional information about diatom seasonality. However, they form an important source of food for foraminifera, as stated in the text, and agree that an additional sentence would be useful, as - broadly speaking - the Chl a pattern does follow what would be expected from seasonal diatom production. We have added om page 4 line 7:*

*"The seasonal progression of Chl a in Antarctic fjords is consistent with a diatom-dominated phytoplankton (Cape et al., 2019; Pike et al., 2008 and references therein; Montes-Hugo et al., 2009)"*

As more general comments:

- I think that the δ13C record could be better interpretetd/used: is there any relation- ships with the chlorophyll maximum ? With the nutrient proxies ? The primary productivity ?

*We have now added to page 8:*

*"There were no significant seasonal differences in variance between samples in the single-specimen $\delta^{13}C_{np}$ dataset (Levene's test, p=0.076), and no links with indicators of primary production."*

- You should discuss the potential impact of the carbonate ion concentration on the shell thickness as well as on the δ18O;

*We have not discussed the impact of carbonate ion concentration on either shell thickness or δ18O due to a lack of data with which to draw comparisons. However, this is a fair point as there are a few studies suggesting that there are subtle changes to both parameters with changing carbonate ion concentration, and we have added the following on page 7:*

*"Whilst temperature alone has no detectable influence, a combination of temperature and pH changes are known to impact calcification rate in both juvenile and adult Nps (Manno et al., 2012). However, as we do not have carbonate ion concentration data available for this location and time period, we will restrict the interpretation of grey value, and so shell thickness, to calcification changes relating to ontogeny."*

*And on page 8:*

*"Note that carbonate ion concentration has only a small impact on foraminiferal $\delta^{18}O$ (~0.002 to 0.004 ‰ per μmol/kg for Globigerina bulloides; Lea et al., 1999) and so cannot explain the range in values observed."*

- What consequences the results of this work can have for paleoclimate studies ? To finish, a very small error appears p7 l18: "peaks" written 2 times.

*Done*

Despite these few comments/suggestions, I think this paper is a very nice contribution to the better understanding of the carbonate productivity in this area, and to the ecological constrains on N. pachyderma.